# Social-Ecological Correlates of Regular Leisure-Time Physical Activity Practice among Adults

**DOI:** 10.3390/ijerph17103619

**Published:** 2020-05-21

**Authors:** Timothy Gustavo Cavazzotto, Enio Ricardo Vaz Ronque, Edgar Ramos Vieira, Marcos Roberto Queiroga, Helio Serassuelo Junior

**Affiliations:** 1Department of Physical Education, Midwestern Parana State University, Guarapuava 85040-167, Brazil; queiroga@unicentro.br; 2Department of Physical Education, State University of Londrina, Londrina 86057-970, Brazil; enioronque@uel.br; 3Department of Physical Therapy, Florida International University, Miami, FL 33179, USA; evieira@fiu.edu; 4Department of Sport Science, State University of Londrina, Londrina 86057-970, Brazil; heliojr@uel.br

**Keywords:** lifestyle, health risk behavior, health policy, environment, public health, social environment, public health surveillance

## Abstract

This study calculated the exposure-response rates of social-ecological correlates of practicing regular (>150 min/week) leisure-time physical activity (PA) in 393,648 adults from the 27 Brazilian state capitals who participated in a national survey between 2006 and 2016. Regular PA encouraging factors were inputted into an exposure-response model. Growth rates for the odds ratio and prevalence of regular PA were calculated for each increase of one encouraging factor. Regular PA was reported by 22% of the participants (25% of men and 20% of women). More than 40% of men and 30% of women with higher intra-personal encouraging conditions reported practicing regular PA. There was a 3% (ages 18–32 years) to 5% (ages 46–60 years) increase in regular PA practice in men for each increase in an encouraging climate factor (temperature from 21 °C to 31 °C, humidity from 65% to 85%, 2430 to 3250 h of sun/year, and from 1560 to 1910 mm of rain/year). Encouraging intra-personal factors and favorable climate conditions had larger effects on regular PA practice than the built environment and socio-political conditions; the latter two had independent effects, but did not have a cumulative effect on PA.

## 1. Introduction

Regular practice of physical activity is associated with primary and secondary prevention of chronic diseases [1], mental illness [2], and premature mortality [3]; it can also significantly reduce the economic burden of these conditions [4]. Although these effects are widely recognized, a planned 10% reduction in worldwide physical inactivity by 2025 is unlikely to be accomplished [5]. To date, science and politics failed to significantly increase regular practice of physical activity in the population. Therefore, new initiatives to monitor and promote regular practice of physical activity are needed [6]. Identifying correlates of healthy and unhealthy behaviors using behavioral theories, such as a social-ecological model, is essential [7,8,9,10]. Thirty-six correlates and 20 determinants of regular physical activity practice have been identified using a social-ecological model. The factors were classified as (I) Environmental—presence of parks and green areas, environment aesthetics, and climate conditions [8,11,12,13,14]; (II) Intra-personal—motivation, age, sex, educational level, ethnicity, weight, perceived exertion, social support [8,13,15,16], genetics [17], and hereditary characteristics [8]; and (III) Social-political—public transport, criminality, legislative or regulatory actions, campaigns, public investments [18], economy, natural disasters, and sport competitions. At the political level, the quantity and quality of evidence are still incipient [8].

The theoretical framework often used in behavioral research relies on models of rational decision-making, where the adoption of a healthy behavior is based on the observation and processing of information for rational decision-making that occurs in a similar fashion for all individuals [19]. However, when human rationality was experimentally evaluated, people made elementary and systematic errors in their reasoning, judgment, and decision-making [19,20]. Rationalized behaviors are influenced by a few simultaneous factors, while non-rationalized behaviors seem to have intuitive, reactive characteristics [19]. Most previous research of factors associated with practicing regular physical activity used theoretical frameworks of rational decision-making. Addition studies using a socio-ecological approach that goes beyond rational decision-making are needed for further understanding and increased predictive ability.

Another issue related with the previous research in the area of the determinants of physical activity practice is that the information on correlates of health-related behaviors come predominantly from research conducted in high-income countries [8]. Several of the studies did not have the specific aim of identifying correlates, reducing the number of factors scanned and overestimating the effects. Furthermore, the analysis of the studies was operationalized for each factor or was performed without adjusting for covariates, therefore not controlling for multicollinearity. Consequently, the effects were observed for each variable independently, with considerable ecological bias [21]. Therefore, the correlates identified to date do not provide enough and adequate support for policy decisions. For that purpose, the creation of comprehensive, adjusted, and real-world compatible models is needed. To design such a comprehensive model, large-scale data on a diversity of factors from different classes and data from mid-income/developing countries are necessary.

Brazil has a large territorial area (8,516,000 km^2^), large cultural diversity, as well as regional differences in climate, behaviors, educational and socio-economic levels, and politics. Since 2006, the Brazilian phone surveillance system (VIGITEL—Risk Factor Surveillance System and Protection for Chronic Noncommunicable Diseases—http://svs.aids.gov.br/bases_vigitel_viva/vigitel.php) has been used annually to gather self-reported information from adults (>18 years) on exposures to risk factors for non-communicable diseases. The data indicate, for example, that physical activity is more frequent in younger adults, men, and in those with higher educational levels. It also indicates that the prevalence of physical inactivity decreased from 16% in 2009 to 14% in 2011 [22]. This progress is possibly related to improvements in strategic plans, increased public and private investments in health, sports, and leisure [22], as well as varying climate conditions during this period.

Comprehensively understanding physical activity requires articulation and advanced methodological models that make it possible to aggregate multiple factors at many scales, namely, biological, social, environmental, and, above all, ecological [8,10]; this in addition to including the temporal, regional, and cultural characteristics to highlight the main predictors of the target behavior within a complex system. Considering the need for large-scale data on a diversity of factors from different classes associated with performing physical activity in mid-income/developing countries, this study used 11 years of social, political, environment, climate, and intra-personal data from VIGITEL to calculate the exposure-response rates of social-ecological correlates for physical activity in adults from the 26 Brazilian state capitals and one federal district. Our hypothesis was that few classes of correlates will have cumulative effects. Intra-personal factors would have a higher effect on physical activity than the built environment. Additionally, the built environment would play a larger role than the other factors for younger men and therefore explain why this group tends to have higher rates of physically activity than other subgroups.

## 2. Methods

### 2.1. Design and Sample

A social-ecological exposure-response analysis of 11 years (2006 to 2016) of pooled data from 572,477 VIGITEL participants from the 27 Brazilian state capitals was conducted. The National Research Ethics Commission (CONEP) of the National Health Council/Ministry of Health approved VIGITEL (No. 13081).

This was a retrospective longitudinal study based on secondary data analysis from multiple databases. The procedures were carried out in two phases: identifying data sources and gathering information, as well as data processing (Figure 1). In the phase of data source identification and gathering, we conducted a search on public platforms and databases. We separated the data into two groups: intra-personal data, composed of data from the VIGITEL survey; and socio-ecological data, composed of information from the 27 cities in the study. We found 116 intra-personal variables and 50 socio-ecological variables. We selected 17 intra-personal variables and 32 social-ecological variables from 27 cities (state capitals + a federal district). The processing phase involves cleaning/imputing, concatenating, and analyzing the data.

We did not include older adults (≥60 years old; in Brazil and in other developing countries, the official classification for “older adults” is age ≥60 years old, as per Art. 1 of Law Nº. 10.741, of 1 October 2003) in this study because we assessed the effects of different variables on leisure-time physical activity. The variables and their medication effects are likely different for older adults and separate studies and models are needed for that population. After excluding incorrect, incomplete, and data from older adults, the final sample consisted of 393,648 adults (18 to 59 years old).

### 2.2. Data Source, Gathering, and Procedures

VIGITEL started in 2006 by the National Health Surveillance Department in collaboration with the Center of Epidemiological Research in Nutrition and Health of the University of São Paulo (NUPENS/USP). The annual results published include data on the frequency, distribution, and trends of the main risk factors for chronic non-communicable diseases in all Brazilian state capitals and from the national capital (Brasília). These cities (Figure 2) represent the largest populations of each state. The sampling procedures aim to obtain probabilistic samples of the adult population (≥18 years of age) residing in households with at least one phone (only regular home phones—landlines; cellphone numbers were not used, as most households in Brazil still have landline telephones). A minimum sample size of approximately two thousand individuals from each city was used to estimate, with 95% confidence and a maximum error of 2% points, the frequency of the main risk factors for chronic non-communicable diseases in the adult population.

The VIGITEL 2006–2016 questionnaires included information on demographics (age, sex, marital status, ethnicity, education level), nutrition (daily and weekly consumption of fruits, vegetables, fat, milk, meat), weight, height, alcohol and current tobacco use, self-rated health, diagnosis of hypertension, and diabetes. Physical activity levels was based on the following questions: “(1) In the last three months during your leisure-time, did you perform any exercise or sport?”; “(2) What is the main type of exercise or sport that you practiced?”; “(3) How many days a week do you usually practice exercise or sport?”; “(4) On these days, how long (minutes/day) do you perform the exercise or sport for?”. The answer options for Question 2 include 15 exercises (walking, treadmill walking, running, treadmill running, resistance training, aerobics, swimming, gymnastics, swimming, martial arts and fighting, cycling, football, basketball, volleyball, tennis, and other exercise). We used the physical activity answers to estimate the weekly dose of physical activity in minutes. Then, we classified the physical activity into two groups using 150 min/week as the cut-off for regular physical activity. All data were analyzed by comparing the groups performing more versus less than 150 min/week of leisure time physical activity.

The data was obtained from different platforms and compiled in a database for the extraction of the factors. All data are available online; the Appendix A describes the data and references (Appendix A).

### 2.3. Intra-Personal Data

The intra-personal variables were classified under the following categories:Demographics = age, marital status, and sex;Health = alcohol abuse, self-related heath, diabetes (medical diagnosis), hypertension (medical diagnosis), obesity (BMI > 30 m/kg^2^); TV time > 3 h/day; and current use of tobacco;Nutrition = regular consumption (≥5 days) in the previous seven days of fruits, meat, milk, soda, and vegetables; and consumption of the fat cap and skin when eating meat and chicken, respectively.

### 2.4. Social-Ecological Data

The socio-ecological data were obtained from the IBGE (Brazilian Institute of Geography and Statistics—www.ibge.gov.br); Government Transparency Portal (www.portaldatransparencia.gov.br); INPE (National Institute of Space Research www.inpe.br); DATA-SUS—SUS (Department of Computer Science of the Unified Public Health System—complex and complete national public system of health care—www.datasus.saude.gov.br/datasus); National Treasury Secretariat (www.tesouro.fazenda.gov.br), and National Department of Transportation (www.infraestrutura.gov.br). Data were obtained for each city and year of the survey. For comparison between cities of different population sizes, we adjusted values to rates/100,000 inhabitants. There were 5% missing data due to the temporal nature of the variables; the missing data were imputed using estimation methods based on seasonal decomposition using the statistical package “imputeTS” in the R software, version 3.5.

The socio-ecological data included the following social, political, environmental, transport, and climate-related variables:Social = crime mortality (1/100,000 inhabitants), number of employees of physical activity-related companies (inhab. rate), family income < 1/2 min wage (%), family income from 1/2 to 1 min wage (%), family income from 1 to 2 min wage (%), family income > 2 min wages (%), percentage of women (%), life expectancy for men, women and in general (years), population (millions), and traffic accident mortality (1/100,000 inhabitants);Political = basic grocery cost (cost of basic food items to feed a family of 4 for 1 month in BR$), availability of clean drinking water (%), family health care teams/primary care teams (n), GDP/capita (BR$), public primary care coverage (%), income inequalities—GINI index (ua), public investment in sports and leisure per capita (BR$), public investment in health care per capita (BR$), and private health insurance (1/100,000 inhabitants).Environment and Transport = bus fleet/100,000 inhabitants, car fleet/100,000 inhabitants, PA companies (e.g., sports and recreational clubs, gyms) (inhabitants’ rate), and vehicle fleet/100,000 inhabitants (all type);Climate = hours of sun/year, max temperature (°C), min temperature (°C), average humidity (%), max humidity (%), min humidity (%), and precipitation/year (mm^3^).

### 2.5. Concatenation and Analysis

The social-ecological data were separated into thirties. The cut-off points for each variable are described in the Appendix A. The social-ecological data were concatenated to the intra-personal correlates/year/city.

### 2.6. Descriptive Analysis

The intra-personal results are presented as relative frequency and confidence intervals. The social-ecological data are described as the average, standard deviation, minimum, and maximum. All analysis was done using the “survey” package in R software and the complex sample model in SPSS 25.0, applying the strata and sample weights provided in the database.

### 2.7. Exposure-Response Model

The multiple binary logistic regression model for complex samples was calculated in SPSS 25.0 with all correlates as possible predictors and covariates. We detected the encouraging conditions for each correlate (Appendix A). The factors were classified as encouraging (OR > 1; P < 0.05) or discouraging/neutral. Each class was created to represent the exposure frequency to encouraging ecological correlates (scenarios). In these scenarios, each person was exposed to a quantity of encouraging factors from each social-ecological class. Thus, we estimated how many encouraging factors each person has been exposed to within each social-ecological class. Subsequently, a new logistic regression model was calculated to determine the odds of PA > 150 min/week with each increase in exposure. The low exposure scenario was used as a reference for all calculations. We used the prevalence (%) and odds ratio (OR) for each scenario to estimate the growth rate (GR_PR_; GR_OR_) for each one encouraging factor increase (the slope in the exposure-response model) using the equation:(1)$$\mathrm{GR}=\frac{\left(x_{2}-x_{1}\right)+\left(x_{3}-x_{2}\right)\ldots \left(x_{i+1}-x_{i}\right)}{n-1},$$
where x_1_ is the value for the first encouraging factor; and n is the number of encouraging factors.

To estimate the cumulative rate according to the difference between the last and first encouraging factor, we applied ∆_PR_ = x_i_–x_1_; considering x1 as the prevalence of the first encouraging factor, and x_i_ the prevalence of the final encouraging factor.

### 2.8. Generalized and Pooled Effects

Finally, we implemented a meta-analysis model using the odds ratios and confidence intervals for the estimation of exposure effects adjusting for age and sex using the “meta” package in R software, version 3.5.

## 3. Results

Twenty two percent of the participants reported practicing more than 150 min/week of physical activity men = 25.3 (24.8; 25.9), and women = 19.5 (19.1; 19.9) (Table 1). The prevalence of obesity and hypertension ranged from 16% to 22%. The prevalence of diabetes was 6%. Alcohol abuse was more than twice more frequent in men than in women. More than 50% of the participants consumed fruits, vegetables, and milk regularly, but men had worse nutritional behaviors than women did. For example, the consumption of meat, meat fat, and chicken skin was approximately twice in men compared to women.

Annual per capita investments in leisure and sports were, on average, R$10.51 (~US$2.5), ranging from BR$0.00 to 117.62 per year (~US$0–27.6). The basic grocery cost ranged from BR$ 132.14 to 459.02 (~US$33–115) depending on the city and year (Table 2). The number of inhabitants per sport/exercise-related business ranged from 650 to 7800. Large variability among cities and periods was also observed in GDP, mortality from crime, and traffic accidents, as well as vehicle fleet. Life expectancy was eight years longer for women (77 years) than for men (69 years).

The results of—were described from an exposure-response model estimated based a preliminary logistic regression model, presented in the Appendix A. In addition, in the Appendix A we present the cut points from which originated the groups (tertiles T1, T2, and T3) of social-ecological correlates. Table 3 shows the social-ecological exposure-response effects on PA > 150 min/week using the growth rate of odds ratio and prevalence change; growth rate of prevalence and odds ratio; and the effect size estimate for each class in the meta-analysis. For both men and women, nutrition and health-related variables had a higher effect on physical activity, especially in younger adults. For men, the inclusion of one encouraging nutritional factor increased the odds of physical activity > 150 min/week by 0.9. Climate had larger and more consistent effects in men than in women; the GR_OR_ for each encouraging factor included in the model was approximately 0.5. The prevalence of regular physical activity increased from 3% to 5% for each encouraging climate-related factor. Interestingly, in women, climate promoted considerable changes in the prevalence of PA only for the younger individuals (18–32 years). The built environment and social class had moderate effects, but these effects were higher in younger men compared with other age groups and to women. In the pooled effects, a higher ecological effect was observed in younger men (Table 3).

The exposure-response effects were not as evident in some classes, but the effects in highly encouraging conditions, especially in younger men, were large and were associated with a rate of physical activity > 150 min/wk of about 40% (Figure 3). Exposure to all encouraging climate conditions promoted an ~8% increase in the prevalence of physical activity in younger men, but they already had a higher prevalence of physical activity without any encouraging climate condition (~30% vs. ≤20% for the other age groups and for women). In women, encouraging nutritional and health-related factors were associated with a more than 10% increase in the rate of physical activity > 150 min/week; encouraging conditions for these classes of factors promoted a physical activity prevalence of more than 20% in younger women. On the other hand, the effects for the other classes were minor or non-existent (Figure 3).

## 4. Discussion

Although previous studies applying social-ecological models have identified similar correlates of physical activity practice [8], individuals are not exposed to isolated social-ecological factors. The effects are dependent on concomitantly/cumulative exposures because people are unlikely to rationalize all social-ecological exposures; and the exposures rarely occur in isolation. If one factor/correlate has an impact on behavior; in theory, two factors should increase and add their independent effects. However, collinearity exists. Our study is innovative because it accounts for this collinearity and evaluates the individual and combined effects on PA practice of different exposures to ecological factors. Previous studies have shown results from independent associations, since they applied multivariate analytical models that control the effect of an association independent of the other related factors. In our study, we adopted an analytical strategy with a cumulative effect. We created predictors that combined encouraging factors. The evaluation of these cumulative effects is the innovation and major contribution of this paper to science and practice. Our study revealed that intrapersonal factors, resulting from cumulative healthy nutritional behaviors and adequate health status, had the highest exposure-response effects of physical activity for both sexes. Besides, encouraging climatic conditions (temperature 21–31 °C, humidity 65%–85%, 2430–3250 h of the sun/year, and precipitation 1560–1910 mm^3^) had considerable and consistent effects on the prevalence of PA among men.

For each healthy nutritional behavior included in the model, the OR of regular physical activity practice increased approximately 0.9 for men and 0.8 for women. The nutritional class had higher effects on women, and the increase varied by age group. For example, for the age group 46–59, women with healthy nutritional behaviors (>4 positive factors) had 2.05 times the odds of performing more than 150 min of physical activity per week than those who had no encouraging factors. Adults who ate healthy foods more habitually tended to exercise more regularly. Despite the inability to infer causality given the cross-sectional nature of the surveys, healthy behaviors were associated with each other independently of the initial factor considered. A study examining whether individuals who change their level of physical activity make corresponding changes in their diet found a reduction in total calorie intake, intake of total fat, saturated fat, protein and cholesterol, high-fat, and high cholesterol foods [23]. Women were more likely than men to decrease their total fat and protein intake (*P* < 0.01), as well as their total calories, saturated fat, and cholesterol (*P* < 0.05) as they increased their physical activity levels. However, changes in physical activity were not associated with changes in eating habits [23]. A recent review showed that interventions led to a lifestyle change, independently of the precursor (physical activity or nutrition) [24]. The combination of positive behaviors is exceptionally beneficial for health [25].

Age, overweight, and sex have an essential role in physical activity engagement [8,13,15,16]. In our study, younger men practiced regular physical activity more often than the other age groups, and more than women. Differences among age groups, and the cumulative effects of combined exposures, were higher in men. However, no smoking, no obesity, no alcohol abuse, no poor self-related health, no hypertension, no diabetes, and no TV > 3 h/day had higher effects on the practice or regular physical activity in women aged > 33 years and in younger men. This fact explains, in part, the differences in the prevalence of regular physical activity by age-group in men and the lack thereof in women.

Women aged 46–59 years presented decreased chances of PA > 150 min/week by an increase in encouraging social domains factors. Our assumption is that social factors do not affect age groups equally. According to the social-ecological model [8], older ages tend to be more influenced by political and environmental factors. This result is reinforced when observing that the environmental effects tend to present differences between the age groups. However, for the environment and transport-related factors, for men the effects were higher between young adults. We believe that this effect is evident because young men tend to use public transport more for study, leisure, and work [26]. Surprisingly, the higher relative number of companies in the sports/physical activity sector was not associated with a higher prevalence of regular physical activity practice. Higher vehicle fleet rates, which favor transportation to and from physical activity sites, were also not associated with higher regular physical activity prevalence. However, increased transport may be associated with reduced traffic safety [4,27,28,29] and increased sedentary time [30,31]. Despite the fact that these variables usually demonstrate significant effects on physical activity (sample references), our results suggest that these correlates may not have a cumulative effect. These findings are supported by other studies. For example, a recent natural experiment using activity monitors did not find a change in the usual patterns (daily steps, time spent performing moderate-to-vigorous physical activity, regular sedentary time) between participants who moved to the East Village sports structure (a legacy of the Olympics and Paralympic games in London-2012) compared to a control group [32]. Another study performed a cost-effectiveness analysis and demonstrated that improving sidewalks in Houston, Texas, USA, was not enough to promote changes in total physical activity [33].

Climate conditions had considerable effects in physical activity practice. The relationship between climate and physical activity has been studied in several aspects, including the effects of temperature change in physical activity practice [34,35,36,37]. In general, warmer climates, but not excessively hot, favor physical activity practice; most results indicate that discouraging weather conditions, such as excessive cold or heat and/or persistent rain, are barriers to regular physical activity practice [34]. Nonetheless, in tropical climate environments, the effects tend to be smaller given the smaller temperature variability [38]. In general, in areas further from the equator, warmer weather favors physical activity practice because of the prevailing low temperatures during most of the year, while in areas closer to the equator, the opposite effect is common with warmer temperatures being associated with less physical activity practice [36,37]. In our study, encouraging climate conditions—temperatures from 21 to 31 °C, relative humidity from 65% to 85%, more hours of sun/year, as well as moderate rainfall—were all associated with a higher prevalence of regular physical activity practice, particularly in men. Experimental studies have found that lower humidity improve cardiorespiratory, thermoregulatory, and perceptual responses during cargo transport [39,40,41]. Additionally, maximum evaporative capacity and heat loss ability gradually decrease as relative humidity increases [40,41]. All these results demonstrate that the estimates of temperature and humidity increase for the coming years due to climate change may negatively affect physical activity practice, particularly in tropical climate countries [37]. In this sense, the built environment, such as parks and green areas, can provide mild climatic conditions and encourage regular physical activity.

Although the results of the present study provide important information, some limitations may affect the generalizability of the findings. One of the limitations is the fact that the data was collected during telephone calls and relied on self-report, which may present memory and reporting bias. Additionally, the lack of information on effort intensity during free-time physical activity (in most of the years investigated) made it impossible to estimate the prevalence of moderate vs. vigorous exercise, an important measure associated with health outcomes. Although studies using TV time use two hours as the cutoff point for the risk of cardiovascular diseases, we adopted TV > 3 h as a criterion for determining the risk of physical activity practice. The outcomes are different. Although the variables of sedentary behavior and physical activity show independent behaviors, during leisure time, TV time and PA are competing.

One of the potential limitations of this study is that the data came from large-city residents. The generalization of the findings may be limited to such populations because the factors may be distinct among rural areas residents. In our study, older adults were excluded from the sample, and this criterion can affect the generalizability of the findings. Considering potential differences regarding the use of leisure-time, medication, and related factors from adults (<60 years), separate models for older adults are needed. Finally, we did not study real-time effects, we observed the usual behavior.

However, our study design and multivariable approach has higher ecological validity. Other strengths of the study include the magnitude/large sample of the research, including results from 11 years of survey data on more than 40 variables from close to 400,000 participants from a mid-income country. The adopted models used combine powerful methodological features and advanced analytical procedures, allowed the prediction of outcomes in a simultaneous system closer to the real world, and has high relevance for science and public policy. For example, although investments in the built environment are the priority incentives for public policy results, our study demonstrated that this class of correlate has no cumulative effects. Thus, investments must be well planned. For further research, our results suggest that the effects of exposure to multiple factors, especially intra-personal, may increase the odds of physical activity practice. Therefore, prospective studies evaluating in real-time the cumulative effects of the exposure-factors can reveal important insight into the components that promote long-term physical activity. Further studies, similarly, are needed to evaluate non-leisure time physical activity, such as physical activity as a mode of transportation (e.g., walking and cycling) and work/labor (e.g., nurses, cleaners, warehouse workers, etc.).

## 5. Conclusions

Encouraging factors in the intra-personal and climate domains had effects that were more substantial in regular physical activity practice for both men and women of all age groups. The increase in encouraging climatic factors promoted more chances of regular physical activity practice. The results were more substantial in younger men and in older women. The mechanisms involving engagement in regular physical activity may have distinct and possibly opposite characteristics when adjusted for age and sex. Favorable built environments, social, and political factors may not have a cumulative effect. Brazilian social and political domains did not corroborate to promote cumulative effects in regular physical activity practice, although they had independent encouraging effects.

## Figures and Tables

**Figure 1 ijerph-17-03619-f001:**
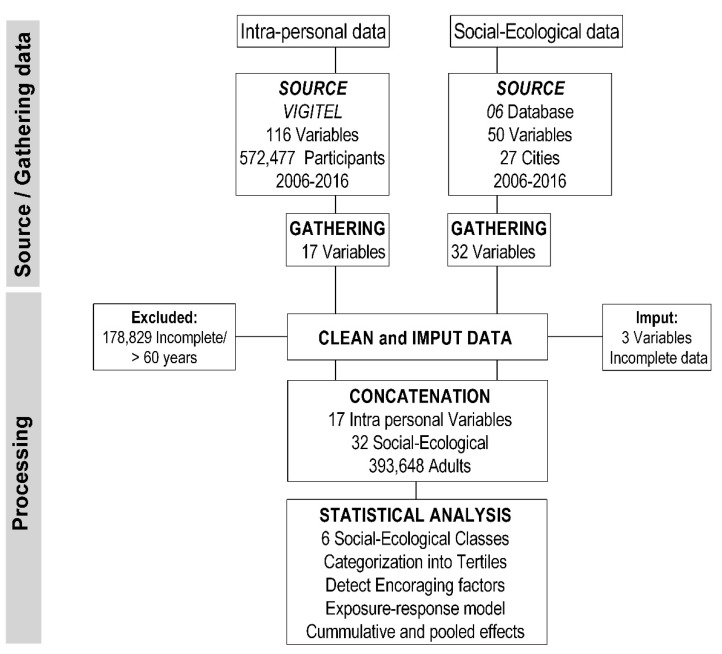
Study flowchart.

**Figure 2 ijerph-17-03619-f002:**
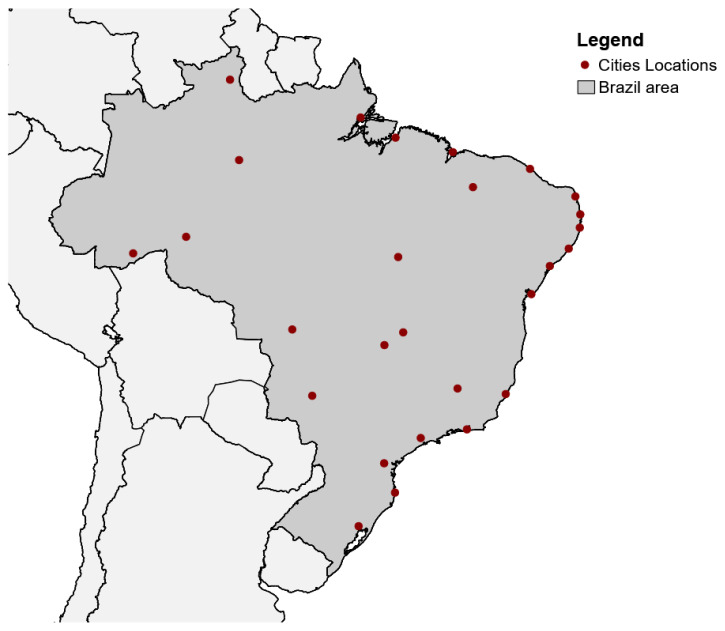
Location of the 27 Brazilian cities in the study. Red dots represent the city locations. The dark gray area represents Brazil, and the light gray area represents other countries in South America that border Brazil.

**Figure 3 ijerph-17-03619-f003:**
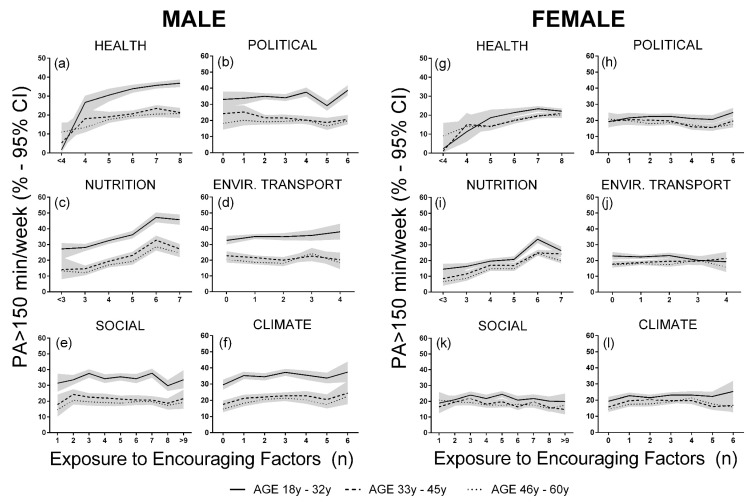
Exposure-response of social-ecological factors on the prevalence of physical activity (PA) > 150 min/week, adjusted by age and sex. (**a**) Exposure-response of health factors on the PA prevalence for men. (**b**) Exposure-response of political factors on the PA prevalence for men. (**c**) Exposure-response of nutrition factors on the PA prevalence for men. (**d**) Exposure-response of environment/transport factors on the PA prevalence for men. (**e**) Exposure-response of social factors on the PA prevalence for men. (**f**) Exposure-response of climate factors on the PA prevalence for men. (**g**) Exposure-response of health factors on the PA prevalence for woman. (**h**) Exposure-response of political factors on the PA prevalence for woman. (**i**) Exposure-response of nutrition factors on the PA prevalence for woman. (**j**) Exposure-response of environment/transport factors on the PA prevalence for woman. (**k**) Exposure-response of social factors on the PA prevalence for woman. (**l**) Exposure-response of climate factors on the PA prevalence for woman. Solid line represents subgroup aged 18–32 years; dashed line represent subgroup aged 33–45 years; dotted line represents subgroup aged 46–60 years.

**Table 1 ijerph-17-03619-t001:** Pooled 2006–2016 self-related descriptive correlates of health, nutrition, and demographic factors by sex.

Classes	Social-Ecological Correlate	All	Male	Female
Outcome	PA > 150 min/week	21.9 (21.6; 22.3)	25.3 (24.8; 25.9) *	19.5 (19.1; 19.9)
Demographic	Age (18–32 y)	31.2 (30.8; 31.5)	34.7 (34.2; 35.3) *	28.6 (28.1; 29.1)
	Age (33–45 y)	31.5 (31.2; 31.9)	31.1 (30.5; 31.6)	31.9 (31.4; 32.4)
	Age (46–60 y)	37.3 (36.9; 37.7)	34.2 (33.6; 34.8)	39.5 (39.0; 40.0) *
	Marital status (Single)	38.8 (38.5; 39.2)	40.4 (39.8; 41.0) *	37.7 (37.2; 38.2)
	Marital status (Married)	40.1 (39.8; 40.5)	41.8 (41.2; 42.4) *	39.0 (38.5; 39.5)
	Marital status (Divorced)	11.6 (11.3; 11.8)	12.7 (12.3; 13.1) *	10.8 (10.5; 11.1)
	Marital status (Widowed)	2.3 (2.2; 2.4)	0.5 (0.4; 0.6)	3.6 (3.4; 3.8) *
	Marital status (Other)	7.2 (7.0; 7.4)	4.7 (4.4; 4.9)	8.9 (8.7; 9.2) *
Nutrition	Vegetables (yes)	53.9 (53.5; 54.3)	48.2 (47.6; 48.9)	58.0 (57.5; 58.5) *
	Fruits (yes)	64.2 (63.8; 64.6)	56.6 (56.0; 57.2)	69.7 (69.2; 70.2) *
	Fruits and vegetables (yes)	39.6 (39.2; 40.0)	32.1 (31.6; 32.7)	45.0 (44.5; 45.5) *
	Meat (yes)	22.2 (21.9; 22.5)	30.6 (30.1; 31.2) *	16.2 (15.8; 16.5)
	Chicken skin (yes)	16.3 (16.0; 16.5)	24.3 (23.8; 24.9) *	10.5 (10.1; 10.8)
	Meat fat (yes)	30.4 (30.0; 30.8)	41.9 (41.3; 42.5) *	22.2 (21.8; 22.6)
	Milk (yes)	53.9 (53.6; 54.3)	57.3 (56.7; 57.9) *	51.5 (51.0; 52.1)
	Soda (yes)	16.7 (16.4; 17.0)	20.3 (19.8; 20.8)*	14.1 (13.8; 14.5)
Health	Weight (kg) ^#^	70.3 (69.7; 70.3)	77.6 (77.5; 77;8) *	65.0 (64.5; 65.3)
	Height (cm) ^#^	165.4 (164.8; 166.0)	172.2 (171.7; 172.5) *	160.6 (159.9; 161.1)
	TV > 3 h/day (yes)	22.3 (21.9; 22.6)	23.3 (22.8; 23.8) *	21.6 (21.1; 22.0)
	Tabaco use (yes)	8.6 (8.4; 8.9)	11.2 (10.9; 11.6) *	6.8 (6.5; 7.0)
	Obesity (yes)	17.6 (17.3; 17.9)	19.2 (18.7; 19.7) *	16.4 (16.1; 16.8)
	Alcohol abuse (yes)	19.4 (19.1; 19.8)	30.1 (29.6; 30.7) *	11.8 (11.5; 12.2)
	Poor self-related health (yes)	3.8 (3.7; 4.0)	2.7 (2.6; 3.0)	4.6 (4.4; 4.8) *
	Hypertension (yes)	20.7 (20.4; 21.0)	19.9 (19.5; 20.4)	21.3 (20.8; 21.7) *
	Diabetes (yes)	5.4 (5.2; 5.6)	5.2 (5.0; 5.5)	5.5 (5.3; 5.8)

Note: Frequency (95% CI) estimated by the complex samples procedure. Nutritional variables refer to regular consumption (>5 days/week). All variables are self-related. * Higher values based on overlapping CI 95%. ^#^ Weight and height were not included in the logistic model.

**Table 2 ijerph-17-03619-t002:** Pooled 2006–2016 descriptive correlates regarding political, environment, social, and climate classes.

Classes	Social-Ecological Correlate	Mean ± SD	Min–Max
Political	Basic grocery package (BR$)	259.1 ± 74.3	132.1–459.0
	Availability of clean drinking water (%)	88.5 ± 12.8	49.2–99.3
	Primary care coverage (%)	63.2 ± 20.3	20.88–100
	Income inequalities—Gini Index (a.u.)	0.5 ± 0.1	0.42–0.60
	Family health care—public health policy (n)	234.1 ± 269.1	41–1531
	Public investment * in sports and leisure per capita (BR$)	10.5 ± 13.26	0–117.6
	Public investment * in health care per capita (BR$)	497.8 ± 227.3	118.2–1260.6
	GDP per capita (BR$)	25,336 ± 13,881	7682–79,099
	Private health insurance (1/100,000 inhabit.)	32,358 ± 14,717	5992–77,525
Social	Life expectancy (years)	72. ± 2.6	67.3–79.1
	Male life Expectancy (years)	69.39 ± 2.6	63.0–75.8
	Female life Expectancy (years)	76.6 ± 2.5	71.2–82.4
	Family income < 1/2 min wage (%)	24.1 ± 10.7	5.2–47.1
	Family income 1 to 2 min wage (%)	18.9 ± 7.9	7.9–43.9
	Family income 1/2 to 1 min wage (%)	24.1 ± 6.0	13.2–38.6
	Family income > 2 min wage (%)	29.3 ± 3.9	18–36.4
	Population (millions)	1721.0 ± 2279.0	0.184–12,038.0
	Traffic accident mortality (1/100,000 inhabit.)	45.3 ± 60.9	0.62–482.3
	Crime mortality (1/100,000 inhabit.)	105.4 ± 163.2	1.05–1098.0
	Number of employees in PA companies (inhabit. rate)	825.2 ± 536.2	194.53–2906.4
	Female proportion (%)	0.5 ± 0.01	0.47–0.54
Environment Transport	PA companies (inhabit. rate)	2888 ± 1561	648–7796
	Car fleet (1/100,000 inhabit.)	386.1 ± 144.6	125.3–740.7
	Bus fleet (1/100,000 inhabit.)	4.3 ± 1.42	1.4–7.9
	Vehicle fleet (1/100,000 inhabit.)	416.2 ± 157.3	135.4–806.5
Climate	Precipitation (mm^3^)	1771.61 ± 591.41	104–3775.6
	Hours of sun/year (hours)	2327 ± 392	475.3–3250
	Max temperature (°C)	30.2 ± 2.7	23.0–35.5
	Min temperature (°C)	21.3 ± 2.9	13.4–25.7
	Average humidity (%)	75.5 ± 6.7	56.2–98.0
	Max humidity (%)	82.8 ± 5.1	62.6–98.0
	Min humidity (%)	66.5 ± 12.6	31.0–98.0

Note: Mean estimated from the public dataset for all 27 cities from 2006 to 2016. SD: Standard Deviation. * Investments funded by city hall. Inhabitant rate: Number of inhabitants for each employee/company. Exchange rate: Jan 2006—1.00 US$ = 2.27 BR$; Dec 2016—1.00 US$ = 3.35 BR$).

**Table 3 ijerph-17-03619-t003:** Effects of exposure to social-ecological classes of variables on the prevalence and odds ratio of physical activity (> 150 min/week) in Brazilian adults by sex and age group.

Social-Ecological Class		AGE 18–32 y				AGE 33–45 y				AGE 46–60 y			
	Sex	Δ_PR_	GR_PR_	GR_OR_	OR (CI95%)	Δ_PR_	GR_PR_	GR_OR_	OR (CI95%)	Δ_PR_	GR_PR_	GR_OR_	OR (CI95%)
**Health**	**M**	35.1	15.1	0.40	1.53 (1.28–1.84)	16.0	5.8	0.10	1.22 (1.06–1.41)	9.8	7.8	0.23	1.39 (1.23–1.57)
	**F**	19.8	12.6	0.41	2.06 (1.50–2.83)	19.9	8.8	0.50	1.37 (1.11–1.70)	11.0	7.1	0.51	1.44 (1.22–1.71)
**Nutrition**	**M**	18.6	17.9	0.91	1.67 (1.25–2.24)	13.1	12.7	0.90	2.01 (1.48–2.75)	11.2	12.2	0.92	2.00 (1.49–2.69)
	**F**	11.8	10.4	0.69	1.79 (1.28–2.50)	15.7	13.3	1.01	2.03 (1.56–2.66)	13.1	11.5	0.82	2.05 (1.52–2.78)
**Social**	**M**	2.2	0.4	0.26	1.22 (1.09–1.38)	2.9	−1.8	0.27	1.40 (1.19–1.64)	5.2	−0.7	0.12	1.29 (1.09–1.51)
	**F**	1.0	−0.8	−0.05	1.18 (1.03–1.34)	0.0	−4.3	0.08	1.23 (1.07–1.40)	−2.7	−2.3	−0.02	0.83 (0.74–0.94)
**Environment** **Transport**	**M**	5.4	3.6	0.31	1.24 (1.09–1.40)	−2.6	−1.6	−0.11	0.99 (0.87–1.12)	−2.2	−0.7	0.12	0.96 (0.84–1.09)
	**F**	−3.7	−3.2	−0.25	0.93 (0.82–1.06)	3.6	2.7	0.08	1.14 (1.02–1.28)	−1.5	−2.1	0.14	1.15 (1.02–1.31)
**Political**	**M**	5.6	5.0	0.04	0.97 (0.89–1.07)	−3.5	−4.4	−0.20	0.79 (0.71–0.89)	1.6	0.1	0.02	0.91 (0.81–1.02)
	**F**	5.7	3.8	0.15	1.14 (1.03–1.26)	0.3	−0.6	−0.15	0.80 (0.71–0.90)	−3.3	−2.5	−0.20	0.82 (0.75–0.91)
**Climate**	**M**	7.9	3.1	0.45	1.33 (1.21–1.46)	6.9	3.7	0.46	1.49 (1.33–1.66)	7.9	4.9	0.50	1.49 (1.34–1.66)
	**F**	6.0	3.3	0.21	1.18 (1.07–1.31)	1.0	−2.3	0.00	1.22 (1.10–1.34)	1.4	−1.2	0.08	1.33 (1.22–1.47)
**Pooled Effect**	**M**	12.5	7.5	0.40	1.29 (1.27–1.37)	5.5	2.4	0.24	1.27 (1.11–1.40)	5.6	3.9	0.32	1.31 (1.15–1.44)
	**F**	6.8	4.4	0.19	1.26 (1.12–1.41)	6.8	2.9	0.25	1.22 (1.07–1.39)	3.0	1.8	0.18	1.16 (1.01–1.32)

Note: Odds ratio estimated by complex samples logistic regression for each correlate and grouped in random-effects meta-analysis adjusted by demographic factors—OR (95% CI). All Classes are a cluster of respective encouraging factors (sum)—M: Male; F: Female; ***Δ_PR_***: Prevalence rate (max–min); *GR_PR_*: Prevalence growth rate; *GR_OR_*: odds ratio growth (interpretation: for each encouraging factor increased in intra-class social-ecological factor, increased *GR (OR* or %) in PA > 150 min/week. Encouraging factors: Health—tobacco use (no), obesity (no), alcohol abuse (no), poor self-related health (no), hypertension (no), diabetes (no), TV > 3 h/day (no); Nutrition—vegetables (yes), fruits (yes), Flv (yes), meat (no), chicken skin (no), meat fat (no), milk (yes), soda (no); Social—crime mortality (T1), traffic accident mortality (T1), population (T1), female proportion (T2), numbers of employees in PA companies (inhab rate) (T3), life expectancy (years) (T3), life expectancy (years) (T3), male life expectancy (years) (T3), female life expectancy (years) (T3), basic grocery package (BR$) (T1), family income < 1/2 min wage (%) (T1), family income 1/2 to 1 min wage (%) (T1), family income 1 to 2 min wages (%) (T1), family income > 2 min wages (%) (T3); Environment—vehicle fleet (1/100,000) (T3), car fleet (1/100,000) (T3), bus fleet (1/100,000) (T3), companies physical activity (inhab rate) (T3). Political: GINI (T1), private health insurance (1/100.000) (T3), GDP per capita (BR$) (T3), FHS public health policy (%) (T3), primary care coverage (%) (T3), public investment in sports and leisure per capita (BR$) (T3), public investment in healthcare per capita (BR$) (T3), drinking water distribution (%) (T3); Climate—hours of sun/year (hours) (T3), precipitation (mm^3^) (T2), max temperature (Celsius) (T2), min temperature (Celsius) (T2), average humidity (%) (T2), min humidity (%) (T2), max humidity (%) (T2). T1, T2, and T3: tertiles, cut-off in detailed (SM2).

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
