# Peer review of "Social-Ecological Correlates of Regular Leisure-Time Physical Activity Practice among Adults"

_ijerph, 2020, doi:10.3390/ijerph17103619_

Round 1
Reviewer 1 Report
This paper stated the association between social-ecological factors and PA. This is not new story in the scientific paper. There are also many concerns that remain to be addressed:
- It is too short for introduction section.
- The authors provided some research about social-ecological factors and PA. However, it would be better if the authors could provide strong argument for discussing previous outcomes in the introduction section.
- Hypothesis is missing.
- The authors must explain the research’s design, not only sample.
- Why authors did not performed multiple binary logistic regression analysis?
- Future directions are missing. I recommend the authors add more information of future studies paragraph in the manuscript.
Reviewer 2 Report
This is an interesting manuscript whose topic is important to be studied. It gives a wide picture of several factors related to physical activity. Please, find my comments and suggestions below.
Page 2, Line 79: Please, clarify why authors considered older adults as ≥60 years old. Usually elderly are considered to be ≥65 years old. Please, add a citation to justify it.
Page 2, Line 88: I assume it was a regular telephone and not a cellular phone. I suggest to specify it in the text.
Page 2, Line 89: I assume the study did not include small towns, so, were there participants from very rural sites included?
Page 3, Line 94: Questions about physical activity levels only include information about exercise and sport, but not about other kind of activities that might be beneficial. For example, gardening, walking, etc.
Page 3, Lines 113-114: Even in page 2, line 93 authors cited that the sugar consumption was considered in the Nutrition information, here it is not included. And later, even in the tables, it is not considered (for example in the note of table 3).
Page 5, Line 176: I suggest to add some information about ethical issues. How they were managed from the National Health Surveillance Department, and how Authors got permission to use these data (even they are public) to develop this study
Page 5, Line 186: I suggest to add in the table 1 the Weight and Height, even the table includes Obesity level related to BMI.
Take care with the table’s Format, as there are some issues that might be improved. For example, Demographic has de las “c” in a different line. In the same table, I suggest that instead of (y) authors put (Yes), as it would be less confusing with the “y” that corresponds to years. I also suggest that, even we have the supplementary data available, it could be good to add the units of each variable. Also, try to include all the data of each variable in one line, as some results numbers are in one line, and others in two.
Page 6, Line 196: Here the same happens with Environment, where the “t” is in a different line. Take care with the “comma (,)” and “dots (.)” that authors have used in the numbers. Sometimes it is confusing, as it is difficult to know when they use decimals or thousands or millions. I suggest to unify tis. For example, when you show the numbers of Hours of sun/years…
Page 8, After the table 2 note: I suggest to explain which kind of supplementary data you are including (without numbers). I mean, tell that data are separated into tertiles (T1, T2, T3), tell that you are using prevalence values (PR)and odds ratios (OR), etc. It will make easier the understanding of tables, even with the notes you put on the initials’ meaning.
Page 9, Note of Table 3: Please, change the format for Climate (it needs to be in italics).
Page 12, Line 29: Please, delete “(Prochaska et al. 2010)”, you have already cited with the reference number.
Page 12-13: concerning Discussion, I think it could be improved a little bit adding some comments about the methods applied in comparison with other previous studies.
Page 13, Line 89: I suggest to add a CONCLUSION subtitle.
Reviewer 3 Report
Generally is well-written manuscript, a few suggestion in attachment

Round 2
Reviewer 1 Report
Now, paper is OK.